# Study on the Utilization of Iron Tailings in Ultra-High-Performance Concrete: Fresh Properties and Compressive Behaviors

**DOI:** 10.3390/ma14174807

**Published:** 2021-08-25

**Authors:** Yunqi Zhao, Xiaowei Gu, Jingping Qiu, Weifeng Zhang, Xiaohui Li

**Affiliations:** 1School of Resources and Civil Engineering, Northeastern University, Shenyang 110819, China; 1810399@stu.neu.edu.cn (Y.Z.); qiujingping@mail.neu.edu.cn (J.Q.); wfzhang@mail.neu.edu.cn (W.Z.); 1910356@stu.neu.edu.cn (X.L.); 2Science and Technology Innovation Center of Smart Water and Resource Environment, Northeastern University, Shenyang 110819, China

**Keywords:** iron ore tailings, ultra high-performance concrete (UHPC), workability, autogenous shrinkage, micro-structure

## Abstract

In this paper iron tailing sand (TS) are used as aggerate to develop ultra high-performance concrete (UHPC). The mix proportion of UHPC is designed and TS were added by 25%, 50%, 75% and 100% (wt.%, i.e., weight percentage) to replace natural river sand. Firstly, the influence of TS on the slurry behavior was carried out. The experimental result indicates that with the continuously increasing content of TS, the workability of slurry decreases, while the air content increases. Considering the workability, the optimal replacing dosage of TS should be less than 50%. Then, tests for the hardened specimens were taken. The compressive behavior and micro-porosity deteriorate with increasing content of TS, and the compressive strength had a positive linear relationship with the workability, which indicated that the decline the compressive behavior is mainly due to the loss of flowability. Finally, autogenous shrinkages of UHPC with different TS dosage were also tested. At the same time, the micro-structure of specimens was discussed, which was deteriorate with the increasing dosage of TS. Therefore, comprehensively considering the compressive behavior, micro-structure and shrinkage behavior, as much as 50% of the aggregate could be replaced by TS when developing UHPC.

## 1. Introduction

Ultra-high-performance concrete (UHPC) is a new type of cement-based material with high mechanical properties (compressive strength and flexural strength of over 120 MPa and 20 MPa, respectively) [1,2,3]. Considering the great mechanical behavior and brilliant performance under high-load condition and disgusting application environment, UHPC is often used in some special field, such as offshore structures, large bridges and special applications [4,5,6]. According to the dense packing model and particle packing model, in a proper particle size range, the closer the material is stacked and the higher the homogeneity of the mixture is, the better the strength and performance of the hardened concrete will be. Therefore, considering that the larger particle size aggregate (such as gravel) is more likely to cause the heterogeneity of the mixture and pores between particles, only fine materials, including cementitious binder and fine silica sand, are usually used to improve the homogeneity and compactness, in order to get a great mechanical and durability properties [7,8,9,10].

Although exhibiting outstanding performance, heave consumption of fine aggregate in UHPC greatly increase the cost and limit the application range. Hence, some researchers have been exploring ways to use other aggregate to reduce the raw material costs in UHPC production. Natural river sand, which exhibits mechanical properties similar to those of expensive silica sands, is widely used aggregate in UHPC [11]. However, the serious environmental problems (such as flooding) caused by indiscriminate extraction seriously limit the using of river sand [12]. In view of this situation, finding a low-cost and qualified material to develop UHPC has become a critical research topic. Investigators have reported using wasted aggregates, such as glass sand [13,14], recycled construction and demolition waste [15,16] and aeolian sand [17] to introduce UHPC. In addition to reducing environmental pressure and production costs, it was also reported that recycled wastes positively affect the properties of the UHPC itself, including its autogenous shrinkage behavior and interfacial transition zone [18].

Iron ore tailing (IT), which is one of the main industrial solid wastes in China, is generated with the beneficiation process of iron ore. The wide use of steel has led to an increase in the deposition of IT. According to the statistics, more than 10 billion tons of TS are discharged every year worldwide, of which storage encroaches upon large landfills [19,20,21]. Therefore, the comprehensive utilization of TS would relieve various environmental and safety risks at the same time [21]. Previous studies have indicated that the chemical composition and particle size of IT is similar to that of fine river sand and generally does not contain harmful substances such as heavy metals. Thus, investigation of using TS in concrete has mainly focused on using as aggregates [22,23,24,25]. However, in order to improve the metal recovery rate, IT is made much finer (less than 1 mm) via a more precise grinding process, which seriously limits the utilization as aggregates. Some scholars have reported using TS as a replacement for cementitious materials. IT mainly acts as filling the internal pores and improving the microstructure of the concrete, which restricts the replacing content due to the low pozzolanic reactivity [26]. To improve the reactivity of IT, physical method (e.g., grinding) is used to transform silica in TS from the quartz condition to an unstable condition [27,28]; however, the pre-treatment approach also increases the cost of the products at the same time. Therefore, considering the properties of low-particle-size and high-quartz-content, preparing UHPC with fine particle size of IT is a good-utilization approach.

Using tailings sand as a replacement of river sand or silicon sand in the preparation of UHPC has been reported in previous studies, where the particle size of tailing sand is much finer than that of river sand or silicon sand, and mainly play a role of pore-filling effect [29,30,31]. However, the shape and angularity of IT show great effect on the flowability and mechanical behavior of concrete [32], so it is necessary to discuss the effect of iron tailings particle morphology on UHPC performance In this paper natural river sand (RS) and iron tailing sand (TS) are used to introduce UHPC individually and in combination, of which the particle size range are similar to avoid the particle size effect. The influence of surface morphology, shape, and mineral composition of the two aggregates on the properties of UHPC are investigated. First, TS is used to replace natural river sand in five proportions (25%, 50%, 75% and 100% by weight) and the properties of fresh mixtures are studied. Then the effect of TS content on the compressive strength of the UHPC is investigated, and its compressive behaviors are discussed in combination with its porosity and interface transition zone. Finally, the autogenous shrinkage is studied to evaluate the impact mechanism of TS on the early properties of UHPC.

## 2. Materials and Methods

### 2.1. Raw Materials

In this study, P.II 52.5 ordinary Portland cement meeting Chinese Standard TB175-2007 and silica fume were utilized as the binary cementitious materials. The chemical compositions and particle size distributions of materials are shown in Table 1 and Figure 1, respectively.

Natural river sand (RS) and iron tailing sand (TS) with the particle sizes of 0.15–0.3 mm were utilized as aggregate to prepare UHPC. TS used in this work were obtained from Benxi City, Liaoning Province, China. Before the test, a sieving experiment was first conducted (shown in Figure 2). The chemical composition and X-ray diffraction (XRD) pattern of the TS were shown in Table 1 and Figure 3, respectively. Both the XRD pattern and chemical composition show that the main component of the TS was quartz. The results show that TS is a typical high silicon quartz-based tailing [27], which is similar to that of the RS.

Scanning electron microscopic (SEM) was used to compare the surfaces and shapes between TS and RS; the images are shown in Figure 4. RS exhibited an approximately elliptical particle shape, whereas TS exhibited an irregular and angular shape. The roundness of two aggregate was analyzed and measured with Image Pro Plus, and the result is also shown in Figure 4. The closer the value of roundness is to 1, the closer the shape of material is similar to a circle. Hence, TS (1.730) has a much more irregular shape than that of RS (1.269). This phenomenon means that, although the particle size of two sands are similar, TS has a larger specific surface area (43.85 m^2^/kg) and greater water absorption (12%) than that of RS (24.10 m^2^/kg and 4%, respectively).

In order to characterize the difference of mechanical properties, a KC-3 particle strength tester is used to measure the particle strength of RS and TS, shown in the Table 2. The result means that tailing sand has a higher particle strength than river sand in a similar particle size range.

### 2.2. Mix Design

The mix design of all of the UHPC mixtures are shown in Table 3, where the water/binder ratio (w/b ratio) used in this study was 0.2 constantly, hence a polycarboxylic based superplasticizer (SP) was used to adjust the workability of the UHPC. T indicates samples to which TS were added as aggregate in UHPC, and the number indicates the extent of replacement (0, 25%, 50%, 75% or 100%).

### 2.3. Mixing Procedure and Sample Preparation

A JJ-5 mixer was used to prepare the mixture, of which the low speed and high speed were 140 ± 5 rpm and 285 ± 5 rpm, respectively. Before mixing, two aggregates were dried at 105 °C for 24 h and then cooled to room temperature. During the cooling process, the container surface was covered to avoid the moisture. The mixing procedure for all of the UHPC mixtures is shown in Figure 5. First, all of the solid materials were added and mixed at a low speed of 140 ± 5 rpm to ensure uniform dispersion. Then approximately 50% of the water was added, followed by the remaining 50% of the water with SP; the resulting mixture was then further mixed at low speed. Finally, all materials were mixed at high speed of 285 ± 5 rpm for another 2 min. After mixing, the fresh materials were cast into molds. Two types of molds with different dimensions were used: 50 mm × 50 mm × 50 mm mold for compressive test [33] based on Standard ASTM C109/C109M-20b and 25 mm × 25 mm × 280 mm mold for autogenous shrinkage test, respectively.

### 2.4. Testing Methods

#### 2.4.1. Fresh Behavior

Workability is an important property of fresh UHPC which significantly influence its mechanical and other properties. After the mixing process, the workability of UHPC was tested firstly, the fresh mixture was poured into a steel cone lying on a flat plate. The cone was lifted vertically while ensuring that the mixture flowed freely. The diameters of the two vertical directions of the mixture were then measured, and the mean of the measurements was used as the workability of the UHPC mixture, as shown in Figure 6.

In order to discuss the influence of TS on the air bubble and pore structure of UHPC, an air content test was carried out using measurement instrument according to the literature [34]. The air content of each fresh mixture was calculated as Equation (1):(1)$$A_{m}=A_{0}-A_{G}$$
where *A_m_* is the air content of mixture, *A*_0_ is the readout of the instrument, and *A_G_* is the correction factor of aggregate.

#### 2.4.2. Compressive Strength

After mixing, all specimens were curing in a water tank with temperature of 60 ± 2 °C for 3 d. The compressive strength was tested and three specimens were carried out for each mixture of which the mean strength value was taken as the compressive strength of the mixture.

#### 2.4.3. Autogenous Shrinkage

A test was carried out to measure the effect of TS content on the autogenous shrinkage of UHPC. After mixing procedure, the slurry was poured into the mold with a size of 25 mm × 25 mm × 280 mm based on ASTM C490-17 [35], and two contactors were embedded at both ends of the specimen simultaneously (Figure 7). After one day under a temperature of 20 ± 2 °C and a relative humidity of 60 ± 5%, the specimen was demolded. Then a contact sensor was used to measure the length and autogenous shrinkage of specimens every 15min, until reach the 7 days. The whole testing procedure is also under a temperature of 20 ± 2 °C and a relative humidity of 60 ± 5% [36].

#### 2.4.4. Pore Structure

A Micromeritics Mercury Porosimeter (AutoPore IV-9500) was used to measure the pore structure of the UHPC, and the test range of pore sizes was 3 nm to 360 μm. Before the test, the samples were soaked in acetone and dried for 4 h in vacuum at 80 ± 2 °C.

#### 2.4.5. Scanning Electron Microscopy Observations

Microstructural studies were conducted using an Ultra-plus SEM (Jena, Germany). Before the test, the sample was covered with gold film. Observations were conducted using a wide range of magnifications from 12× to 1M×.

## 3. Results and Discussion

### 3.1. Workability and Air Content

Figure 8 shows the effect of TS dosage on the workability of UHPC. Compared with reference T0, workability of T25, T50, T75 and T100 decreased 5.67%, 14.54%, 24.82%, and 31.56%, respectively. A similar result was obtained from the experimental study of Mirza et al. [23] and Tian [25]. The poor workability of UHPC incorporating iron tailing sand can be attributed to the coupling effect of paste film thickness (PFT) and water film thickness (WFT) [37]. The slurry formed by fine particles and water fills the voids between the coarse particles, and the excess slurry is wrapped on the surface of the coarse particles to form a paste film [38]. WFT is the thickness of the water film formed by the remaining water wrapped on the surface of the particles after the total water of the system filled in the voids of the solid particles [39]. The replacement of TS leads to a decrease in the overall packing density of the aggregate system. Although the particle size distribution range of TS and RS is similar (150–300 μm), the width of the particle size distribution of TS is significantly narrow than that of RS, which is the main reason for the decrease in packing density [40]. In the case of a fixed volume of cement paste, the decrease in packing density leads to a decrease in PFT, which means that more cement paste is needed to achieve the same workability [38]; at the same time, according to the excess layer thickness theory [41], the highwater absorption of TS leads to an increase in the content of onset of flow water [42]. Therefore, the WFT that is available for increasing the workability must be reduced. In addition, it is worth noting that the friction generated by the irregular and angular shape of TS (shown in Figure 4) make it harder for tailings particles to slide past one another during shearing [43,44].

On the contrast, the air content increase with dosage of TS, as shown in Figure 8. This result means that the addition of tailing sand has a positive promotion on the number of micro-pores in UHPC specimen. The surface of TS was covered by a layer of water due to the higher water absorption ratio, and lead to a greater absorption capacity of air. At the same time, compared with river sand, the particle shape of tailing sand is more irregular (Figure 4), which may lead to more voids due to the internal accumulation structure and increase the air content of mixture [43].

### 3.2. Pore Structure

Figure 9 shows the effect of TS dosage on the micro-pore structure of UHPC after curing. The pore particle sizes of all of the mixtures were mainly within the range 19.93–23.41 nm (Figure 9a). According to a previous study, pores with a diameter of smaller than 20 nm can be considered harmless pores [45]. Thus, we concluded that the pores do not adversely affect the strength of UHPC and that the incorporation of TS does not harm the pore structure of UHPC.

Figure 9b shows that, with the continuously increasing TS content, the number of micro-pores increased. The porosity enhanced from 6.36% to 7.29%, 7.69%, 8.09% and 8.53% for T0, T25, T50, T75 and T100, respectively. The result indicates that the existence of TS as aggregate has a negative effect on the packing model of UHPC. We attribute the deterioration of pore structure in UHPC to two aspects:

(a) the lower workability with the increasing TS dosage will improve the entrapped air and the internal porosity of slurry [30], and finally leads to the growth of micro-pore, which was also consistent with the results of spread flow and air content discussed in Section 3.1;

(b) the different shape between two aggregate also influenced dramatically the packing model of UHPC. Ostrowski et al. discussed the morphology of aggregate on the properties of HPC (high-performance concrete), of which the result is similar to that of this study [46,47]. Elliptical shape of RS (Figure 4b) makes the particles compact close, and the smaller particles can effectively fill the holes between the larger particles (Figure 10a). This close contact and filling effect between aggregates can make the packing model more compact and reduce number and size of pores. Nevertheless, compared with RS, TS is much more angular and presents the shape of concave polyhedron (Figure 4a). This shape feature prevents the aggregate from contacting closely with each other (Figure 10b), leading to the increase of porosity [46]. At the same time, the angular shape of TS has a poor filling effect between particles, which also deteriorates the packing model and internal pore structure.

### 3.3. Compressive Strength

Figure 11 shows the compressive strengths of the UHPC samples with different TS contents after curing. With increasing TS content, the compressive strength of the UHPC decreased by 2.05%. 5.61%, 9.50%, and 16.17%, respectively, and the magnitude of the decrease increased with TS content. As reported in previous studies [29,48], due to the finer particle size compared with river or silicon sand, iron tailing in concrete mainly played an actor as filling material, which can slightly enhance the compressive strength due to the much denser structure. However, in this study the particle size of two aggregates are similar. Hence, the filling effect is negligible. As shown in Table 2, although the particle strength of Tailing Sand is slightly higher than that of River Sand, the compressive strength of specimen with TS is still low (especially T100). The reduction of compressive strength can be mainly due to the loss of workability. As discussed in Section 3.1 and testing result in Figure 7, with the increasing replacement of TS, the spread flow diameter (i.e., workability) of slurry decreases sharply. At the same time, the low workability obviously affects the mechanical behavior of hardened UHPC. Figure 12 shows the relationship between the spread diameter (i.e., workability) and compressive strength of UHPC with different TS content. It is obvious that there is a strong correlation between workability and compressive strength. With the increasing content of TS, the workability sharply decreased, finally leading to the loss of compressive strength. On the other hand, the increase of air bubbles (Figure 8) makes the internal structure of UHPC more porous and has an adverse effect on the compressive strength at the same time.

### 3.4. Autogenous Shrinkage

Autogenous shrinkage is a critical factor that affects the mechanical properties and durability of other cement-based materials. Compared with other cement-based concrete, UHPC exhibits greater early autogenous shrinkage due to the low w/b ratio and the substantial amount of binder, which may cause a high internal stress and lead to the generation of microcrack significantly [49,50,51]. Figure 13 shows the effect of TS content on the autogenous shrinkage of UHPC. All specimens show a similar trend in which shrinkage occurs mainly within 1 d, consistent with the previous study [51]. Compared with the reference UHPC sample T0, all the mixtures with TS have a greater 7 d autogenous shrinkage, where the shrinkage increases by 0.63%, 1.75%, 3.50%, and 7.96% for T25, T50, T75, and T100, respectively. This increase of shrinkage value can be explained as that TS absorbs much more water than RS due to the much larger water absorption ratio. According to previous studies, the water-absorption characteristics of aggregates affect the autogenous shrinkage of concrete and a higher water-absorption ratio results in greater autogenous shrinkage [22,52]. The iron ore tailings absorb most free water in the early stage, which leads to the decrease of the internal pressure and the increase of the shrinkage finally. Previous study has also indicated that a finer aggregate particles size increases the autogenous shrinkage of concrete [53]. Although the particle size range of TS is similar to that of natural river sand, considering the complex surface morphology of TS (shown in Figure 4), the shape of tailing sand could be imagined as a finer particle with irregular-shape granule cover. This makes the particle size of tailing sand lower than that of natural river sand, which also has a promoting on the autogenous shrinkage.

It is worth noting that, compared with the reference T0, the greatest autogenous shrinkage increasing of UHPC is 7.96% of T100, which means that the negative effect of TS on the autogenous shrinkage of UHPC is limited.

### 3.5. Microstructure

A micrograph of UHPC is shown in Figure 14. The role of RS and TS in UHPC is similar, both of which bear capacity under compressive loading. It was worth-noting that the surface of TS shows a greater number of scratches than RS. This result indicates that it is more difficult to separate Tailing sand from cement paste, which may be due to the irregular shape. However, a certain width was observed between the two aggregates and paste, referred to as the interfacial transition zone (ITZ). The ITZ is considered the weakest part of the internal microstructure of UHPC because of the precipitation of calcium hydroxide (CH) crystals on the grains [54]. The width of the ITZ of the natural river sand and TS is similar, but the specific surface area of the TS is larger (for the same particle size); thus, the posed by the ITZ is greater, which is also consistent with the result of compressive behavior.

## 4. Potential Application

In this paper, a green-designed UHPC is prepared using iron tailing sand as aggregate. With the rapid development of construction industry and highway transportation industry, the demand for UHPC is also increasing, which means that more and more fine aggregate is required. As a typical solid waste, iron tailing has high potential application value. As a type of aggregate used in UHPC production, it can not only produce economic benefits as a product, but also effectively reduce the exploitation of natural sand and solve the environmental pollution caused by the stacking of iron tailings at the same time.

## 5. Conclusions

This study addresses the properties of a new UHPC incorporating iron ore tailings. Based on the experimental results, the following conclusions are drawn:

(1) With increasing TS content of UHPC, the workability decreases, while the air content increases correspondingly. This result is mainly attributed to the higher water absorption and larger shear resistance of TS than RS during the mixing process.

(2) The compressive strengths of UHPC decrease with the enhancing content of TS, but still reaches 120 MPa with 50% TS content, which is tightly related to the workability of slurry. Therefore, to ensure the workability and mechanical behavior of UHPC, the TS content of should be 50% or less;

(3) The addition of TS has a certain adverse effect on the internal pore structure of UHPC. The angular shape of TS leads to the loose contact between particles, resulting in the increase of internal porosity, and affects the fresh and mechanical behavior.

(4) The autogenous shrinkage of UHPC increases with the addition of TS but is still under 10%. The main reasons are the higher water absorption and finer equivalent particle size of TS. Future research will focus on how to reduce the autogenous shrinkage of UHPC and improve its properties.

## Figures and Tables

**Figure 1 materials-14-04807-f001:**
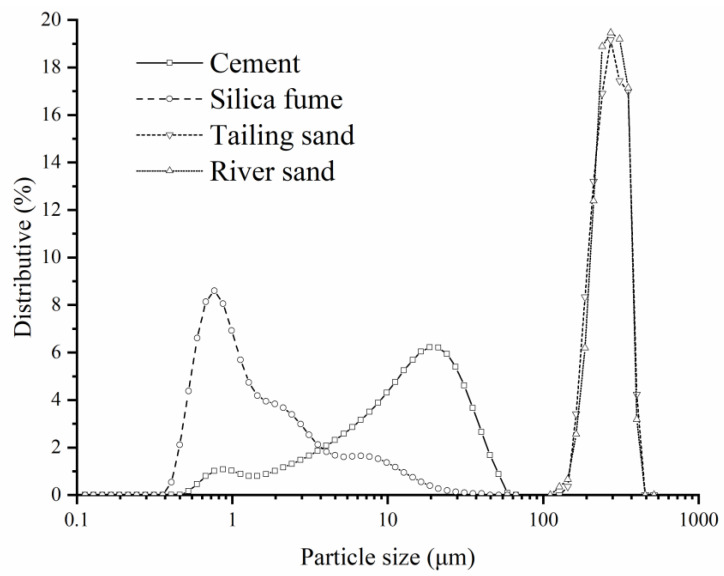
Particle size distribution of binders used in this study.

**Figure 2 materials-14-04807-f002:**
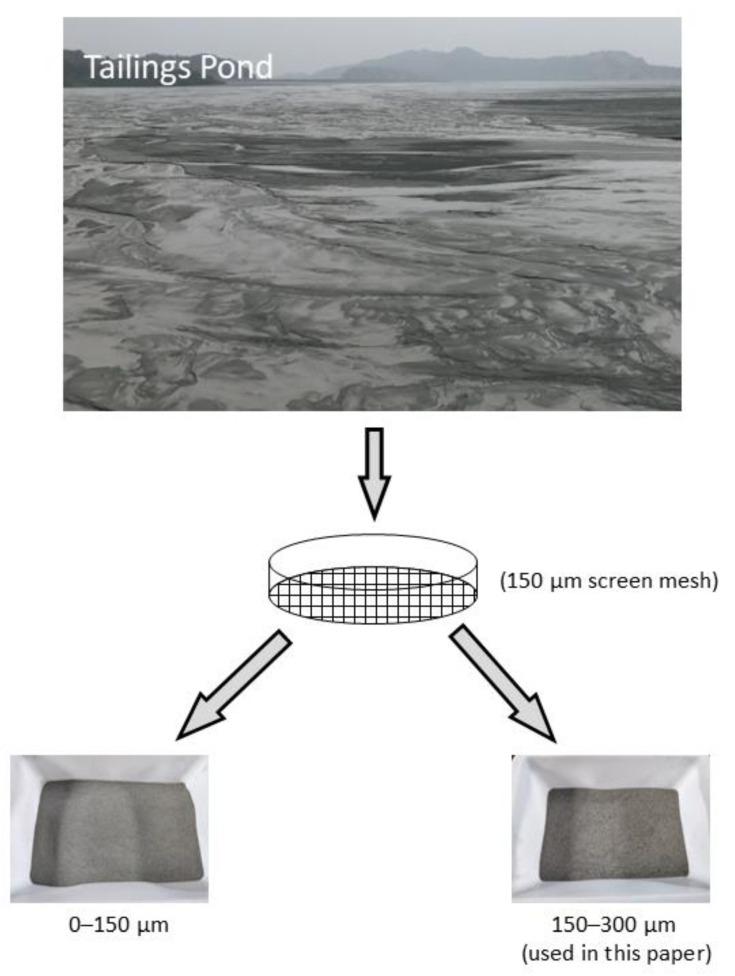
Sieving process of iron tailings.

**Figure 3 materials-14-04807-f003:**
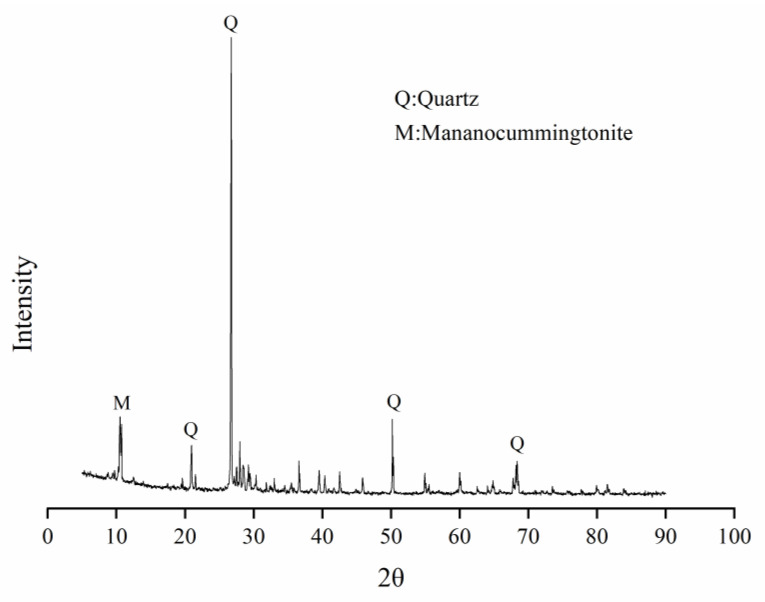
X-Ray diffraction (XRD) patterns of iron tailings.

**Figure 4 materials-14-04807-f004:**
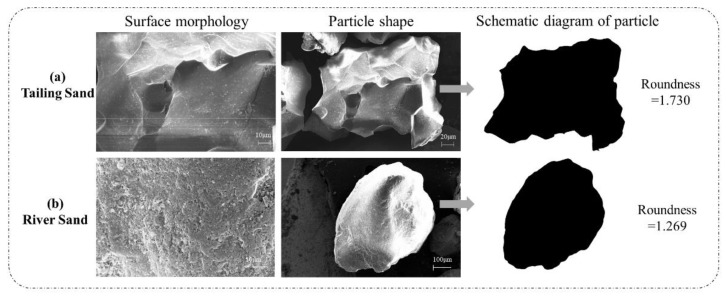
SEM images of iron tailings (**a**) and natural river sand (**b**).

**Figure 5 materials-14-04807-f005:**
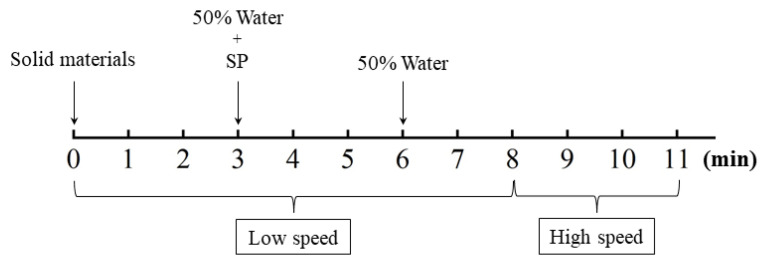
Mixing procedure of UHPC.

**Figure 6 materials-14-04807-f006:**
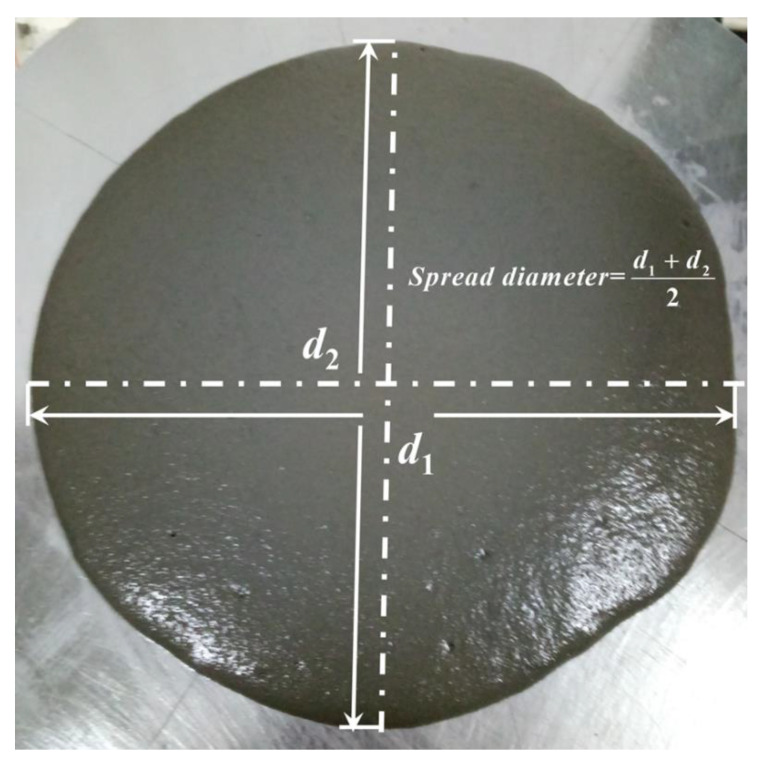
Schematic diagram of workability test.

**Figure 7 materials-14-04807-f007:**
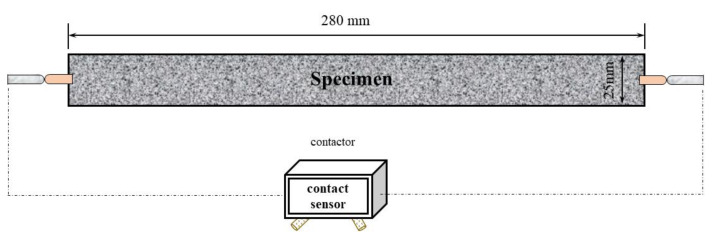
Schematic diagram of autogenous test.

**Figure 8 materials-14-04807-f008:**
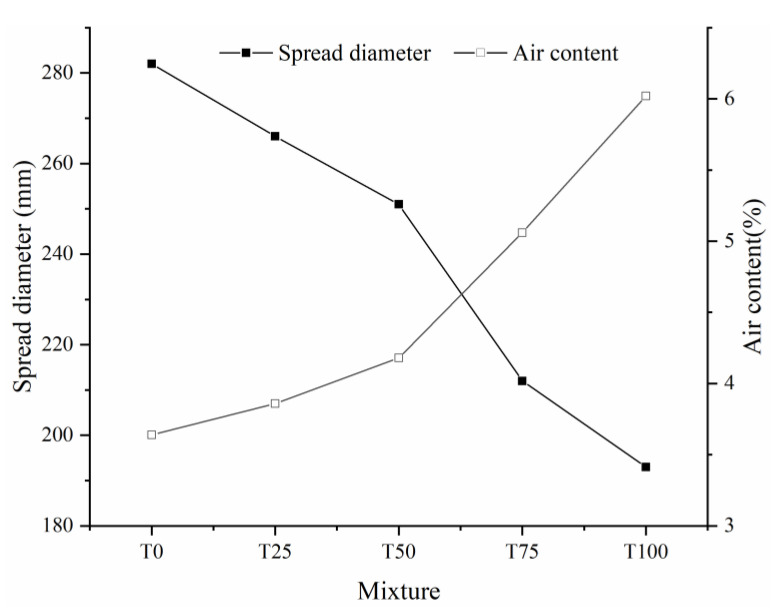
Effect of iron tailings on the workability and air content of UHPC.

**Figure 9 materials-14-04807-f009:**
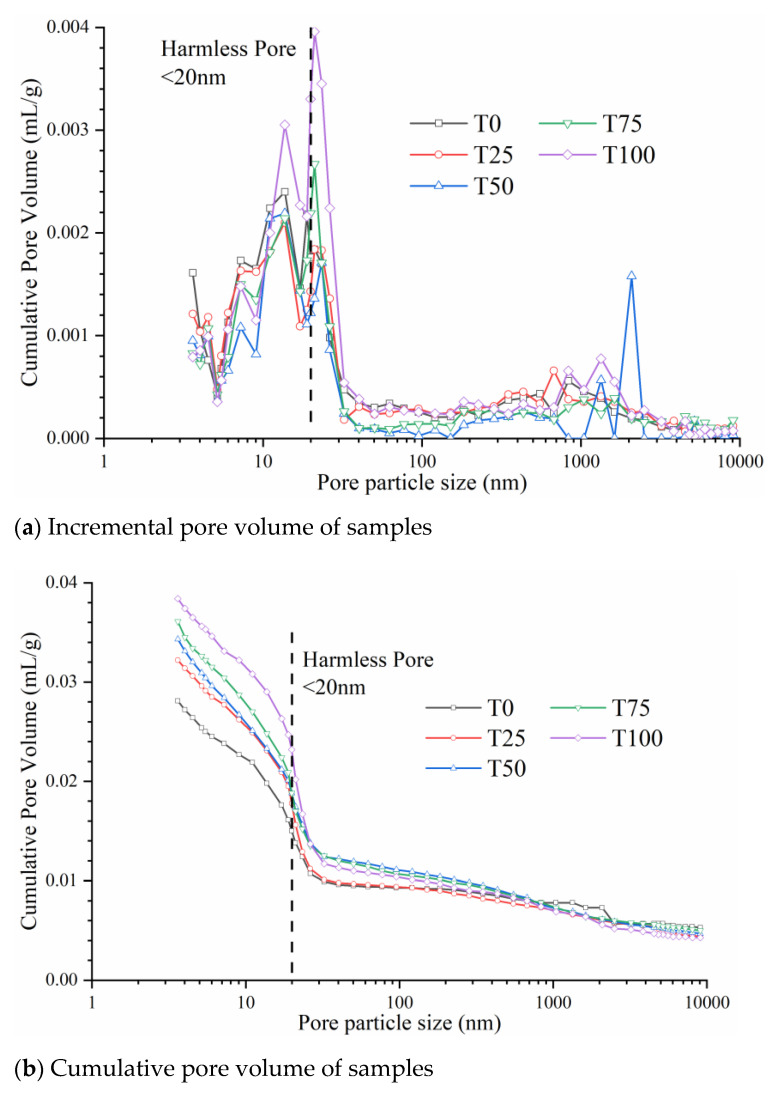
Pore structure analysis of hardened UHPC.

**Figure 10 materials-14-04807-f010:**
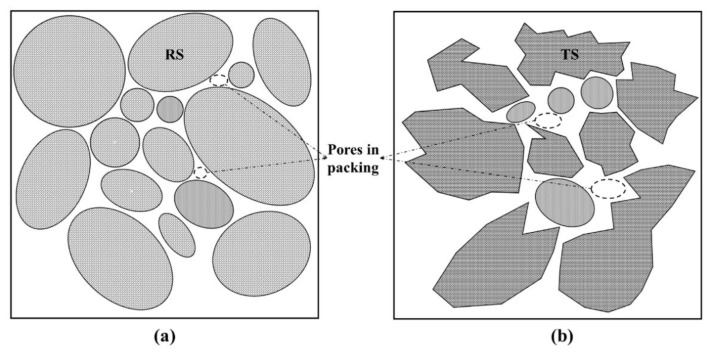
Pattern of particles packing model of two aggregates. (**a**) river sand (**b**) iron tailing sand.

**Figure 11 materials-14-04807-f011:**
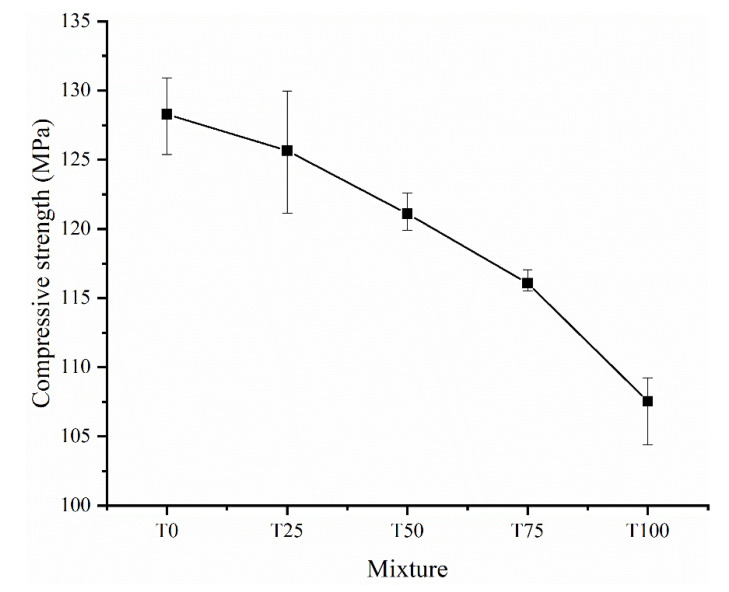
Effect of iron tailings on the compressive strength of UHPC.

**Figure 12 materials-14-04807-f012:**
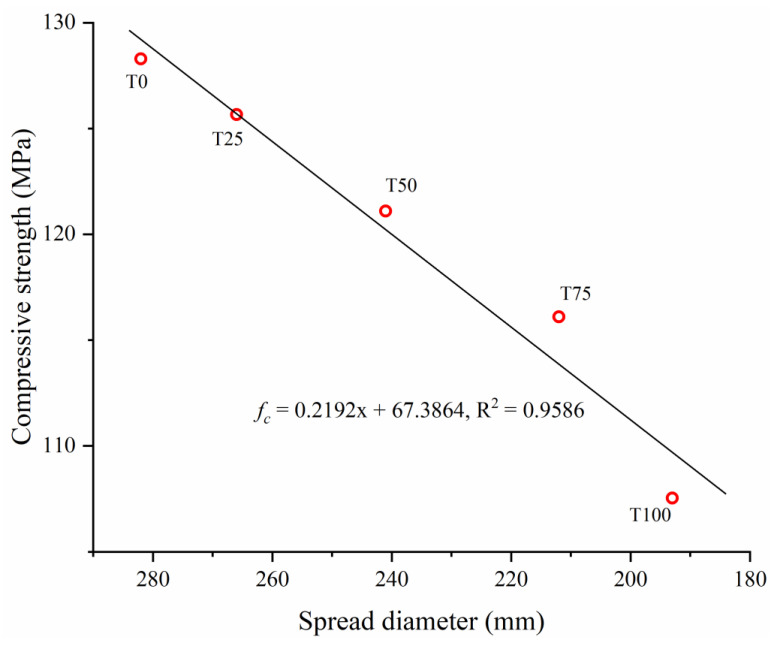
Relationship between workability and compressive strength of UHPC.

**Figure 13 materials-14-04807-f013:**
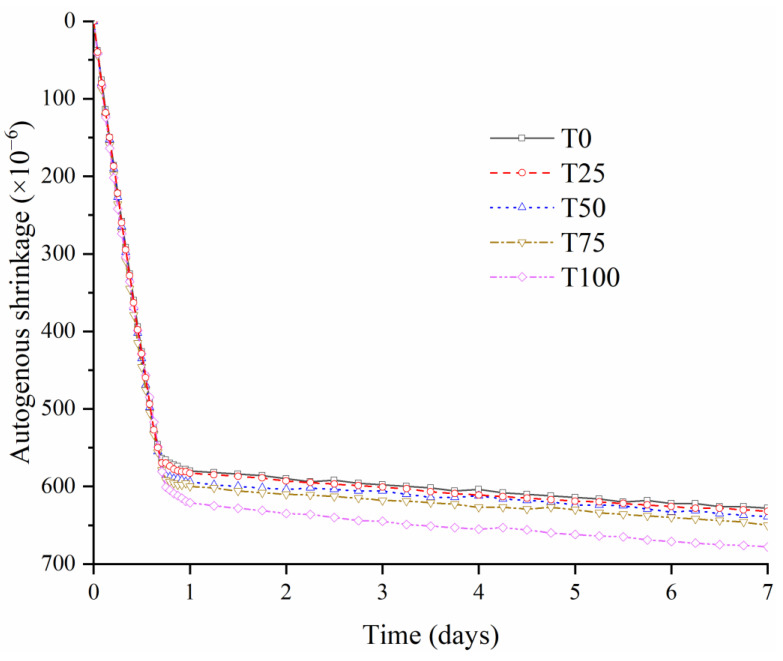
Effect of iron tailings on the autogenous shrinkage of UHPC.

**Figure 14 materials-14-04807-f014:**
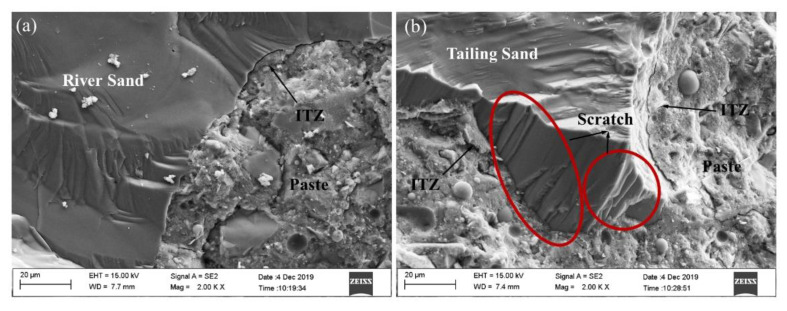
SEM images of ITZ. (**a**) river sand (**b**) iron tailing sand.

**Table 1 materials-14-04807-t001:** Chemical analysis of the cementitious materials and iron tailings (%).

Compositions	Cement	Silica Fume	Iron Tailing Sand (TS)
SiO_2_	20.19	94.77	69.08
CaO	63.03	-	5.05
Al_2_O_3_	5.11	0.35	4.74
MgO	1.72	-	6.06
Na_2_O	0.10	-	0.39
K_2_O	0.32	-	0.34
SO_3_	1.19	-	0.48
Fe_2_O_3_	2.11	-	8.88
LOI	2.14	0.66	0.89

**Table 2 materials-14-04807-t002:** The particle strength of two aggregate.

	1 *	2 *	3 *	Average *
Tailing Sand	98.22	97.99	98.75	98.32
River Sand	96.51	95.89	97.03	96.48

* unit: %.

**Table 3 materials-14-04807-t003:** Mix design of UHPC (kg/m^3^).

	C	SF	RS	TS	Water	SP
T0	780	188	1100	0	194	24
T25	780	188	825	275	194	24
T50	780	188	550	550	194	24
T75	780	188	275	825	194	24
T100	780	188	0	1100	194	24

C: Cement, SF: Silica Fume, RS: River Sand, TS: Tailing Sand, Water: including the water in superplasticizer, SP: superplasticizer.

## Data Availability

Data is contained within the article.

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
