# Peer review of "Study on the Utilization of Iron Tailings in Ultra-High-Performance Concrete: Fresh Properties and Compressive Behaviors"

_materials, 2021, doi:10.3390/ma14174807_

Round 1

Reviewer 1 Report

the article is well written and interesting. I only see a few formatting issues. Pass a line before each figure or table Do not put the title of the figure in bold Put the title of the table just after the number: Table 1 xxxxxxxx Figure 5 is misplaced, it overflows the page. line 249, title of the figure misplaced. line 307, put a line before Author Contribution

after proofreading, part 2 can be improved. indeed, are the shapes of mortars produced based on a standard, why such a quantity of cement? a photo of a workability test is missing to visualize the spread. Why are the specimens stored at a temperature of 60 ± 2 ° C for 3 days, missing information on the date of the compression test, 3 days, 7 days? the figure 5 is too large

Author Response

Answer:

Thanks for your kind suggestion and approval to our study. According to your kind comments, we have modified all the formatting issues on the Figures or Tables, which are marked in yellow background. The oversized Figure 5. is also lessened.

Line 249. The title of figure has been added (now Line 296).

Line 307. The line is added before the Author contribution. (now Line 367)

The shape of mortars in this study is cube of 50 mm × 50 mm × 50 mm, which is usually used in the study on the compressive properties of UHPC or other mortars such as “Wu, Z.;  Khayat, K. H.; Shi, C., Changes in rheology and mechanical properties of ultra-high performance concrete with silica fume content. Cement and Concrete Research 2019, 123, 105786. https://doi.org/10.1016/j.cemconres.2019.105786” and “Zhong, H.; Zhang, M., Effect of recycled tyre polymer fibre on engineering properties of sustainable strain hardening geopolymer composites. Cement and Concrete Composites 2021, 122, 104167. https://doi.org/10.1016/j.cemconcomp.2021.104167”. The following standard ASTM C109/C109M-20b has been added in the manuscript (now Line 157).

In order to achieve a higher mechanical behavior, a large amount (usually from 700kg/m3 to 800kg/m3) of cement are needed.

The photo of a workability test is added to better illustrate the workability test (now Line 171-172).

In order to evaluate the effect of tailing sand on the compressive strength, all the UHPC specimens are cured in a temperature of 60 ± 2℃, which can always be seen in the research about UHPC, such as “Shen P,  Lu L ,  He Y , et al. The effect of curing regimes on the mechanical properties, nano-mechanical properties and microstructure of ultra-high performance concrete[J]. Cement and Concrete Research, 2019, 118:1-13. https://doi.org/10.1016/j.cemconres.2019.01.004”. And a curing time of 48h or 72h is always considered. Hence, in this study 72h is used to ensure the great compressive strength.

Figure 5 has been lessened (now Line 160-161).

Thanks for your kind suggestions very much.

Reviewer 2 Report

This paper investigates the performance of iron ore tailing as a replacement for fine sand in UHPC. The topic is not new. Multiple research already exists in the area. A good example is, https://doi.org/10.1016/j.conbuildmat.2013.10.019, a similar work on UHPC with iron tailing from 2014. All the investigated parameters have already been investigated before, hence this paper does not add a significant contribution. To be of value, the paper needs to include more parameters which are not been investigated before and the analysis and discussions also need to be improved. More comments on the paper can be found in the annotated pdf.

Author Response

Answer:

Thanks for your kind suggestion and approval to our study. Exactly, the preparation of UHPC from iron tailings as aggregate is not such a new topic. Nevertheless, the existing studies have used small-size iron tailings to replace large-size river sand or quartz sand, which means tailings sand mainly play a role of filling-effect material, but the difference of shape between two aggregate is ignored. In this study the effect of aggregate shape on the workability, compressive strength and shrinkage behavior are mainly discussed. Hence, firstly the particle sizes of two aggregates should be same or similar to avoid the filling-effect.

According to your suggestion, we have corrected the mistakes in the manuscript with the yellow background.

Thanks for your kind suggestions very much.

Reviewer 3 Report

The article discuss the topic of the study on the utilization of iron tailings in ultra-high-performance concrete: fresh and compressive behaviors.
The article presents valuable content. In my opinion article should be improved before potential publication.
The following modification should be considered:

1. I suggest to change part of title as follows: Study on the utilization of iron tailings in ultra-high-performance concrete (UHPC): fresh properties and compressive behavior.
2. Abstract line 13 - please explain abbreviation 'wt'.
3. Introduction part is little short and should be rewritten. Consider to add more literature related to the essence of this studies. Please deeply describe positions in literature: 4-10. Moreover, it is recommended to study how geometry of materials used in production of concrete mixture (especially iron tailings and aggregates) could affect on the air-void distribution and compressive strength of high strength concretes. The following literature could be helpful: https://doi.org/10.3390/ma11081372; https://doi.org/10.1016/j.conbuildmat.2019.117794; 
https://doi.org/10.1155/2016/8606505.
4. I suggest to add separated point - Research significance - Please describe here the main essence of the research. 
What was the inspiration for such an analysis? Why presented studies are so important?
5. Figure 3 - please show vertical axis with description.
6. Please improve the quality of figures 4 and 5.
7. How was measured water absorption of TS?
8. Table 2 - please define once again TS, RS and SP.
9. It is strongly recommended to show particle size distribution of RS and TS.
10. Figure 8 - did you determine the morphology of TS and RS? Especially aspect ratio, area ratio, roundness etc?
11. It is recommended to indicate potential application of research results in civil engineering or another discipline.
12. Line 298 - please improve Mpa to MPa.

Author Response

  1. Thanks for your kind suggestion. We have improved the title of our study as your suggestion (now Line 2-3).
  2. Thanks for your kind suggestion. We have added an explanation of ‘wt%’ in the manuscript (now Line 13).
  3. Thanks for your kind suggestion. We have improved the Introduction following your kind suggestion with a yellow background. The three researches you provided are of great reference value, and we have added them to appropriate places in the manuscript and summarized them, which is also marked with yellow background (now Line 31-41 and 74-77).
  4. Thanks for your kind suggestion. We have added ‘4 Potential application’ to indicate the inspiration and importance of this study (now Line 342).
  5. Thanks for your kind suggestion. Figure 3 is a schematic diagram of X-ray Diffraction of iron tailing, which is usually without a vertical axis as follows:

Figure from: https://doi.org/10.1016/j.jclepro.2020.121231

Figure from: https://doi.org/10.1016/j.jclepro.2020.121112

  1. Thanks for your kind suggestion. We have improved the quantity of two figures to make them easier to understand (now Line 126 and 160).
  2. Thanks for your question. The water absorption of TS is determined as follow process: Firstly, TS is placed in a drying oven at 105 ℃ for 24 h to ensure absolute drying, and the weight is measured (G). Then TS is soaked in 20℃ water for 48 h to ensure absorbing water fully, and taken out. A suitable gauze is used to filter out the redundant water in the TS, of which the weight is B. The water absorption of TS is calculated as follow formula:
  3. Thanks for your kind suggestion very much, and we have defined the abbreviation once again (now Line 144 and 145).
  4. Thanks for your kind suggestion very much, and we have added the particle size distribution of RS and TS in Figure 1 (now Line 96).
  5. Thanks for your kind suggestion very much, and we have analyzed and calculated the roundness of RS and TS in Figure 4 (now Line 126).
  6. Thanks for your kind suggestion very much, and we have discussed the potential application of tailing-based UHPC in 4 Potential application (now Line 342).
  7. Thanks for your kind suggestion very much, and we have corrected the mistake (now Line 357).

Thanks for your kind suggestions very much.

Reviewer 4 Report

General Comments:

This paper deals with the experimental study on the properties of a new ultra-high-performance concrete: (UHPC) incorporating iron ore tailing sand (TS). Especially it is considered that the originality is utilization of iron ore tailing sand (TS) for mixing UHPC as substitute materials of fine aggregates. Reviewer considered that the completeness of this paper is very higher. Then this paper provides valuable data for readers.

However it is considered that there are several points that author should make modifications or correction.

1) Line 92 to 93:

The meaning of “Fig. 2 Sieving process of iron tailings” is incomprehensive for reviewer. Author should explain the meaning a little in detail.

2) Line 93 to 97:

There is no information of the physical properties or mechanical properties on Natural river sand (RS) and iron tailing sand (TS). Should describe the new table for the mechanical properties of fine aggregate used in the experiments. Also, should describe the data of cement and silica fume used.

3) Line 122:

Reviewer cannot understand the “JJ-5 mixer”well. Should explain “JJ-5 mixer” a little.

4) At Fig. 5 Mixing procedure of UHPC:

There is not a unit. Should add a unit.

5) Line 131:

“50x50x50mm3” should revise in “50mm x 50mm x 50mm”.

There are some similar modified points elsewhere.

6) Line 144 to 149:

The meaning of this part is incomprehensive for reviewer.

In this experiment, did authors use the AE agent for this concrete (UHPC)?

Should explain a little in detail.

7) Line 168:

“from 100 × to 1200 ×” is wrong. Please revise it.

8) Line 170 to 193:

The meaning of this part is incomprehensive for reviewer.

In Fig.6, it is considered that we might control spread diameter is constant if we could adjust addition quantity of the SP? Please explain the reason that unit content of SP are “24kg/m3” constantly in the Table 2.

9) Fig. 11:

Should describe the explanation of the graph which spread in the Fig.11 in the paper.. It's so complicated.

Therefore this paper is required to be improved to be published in the in the Journal of Materials.

Author Response

Answer:

Thanks for your kind suggestion and approval to our study. According to your kind comments, we revised some details in the manuscript to make it easier for readers to read. And the detailed modifications are as follows:

1) Thanks for your kind suggestion. We have improved Figure. 2 and added more details to make it easier for readers to understand (now Line 110).

2) Thanks for your kind suggestion. We have conducted a particle strength test to characterize the mechanical properties of the two aggregates. Relevant results and analysis have been added in the manuscript (now Line 129-135).

3) Thanks for your kind suggestion. The JJ-5 mixer is a kind of equipment for mortar mixing, while “JJ-5” is the model of this equipment, which is widely used in UHPC preparing, such as “Zhao S ,  Fan J ,  Wei S . Utilization of iron ore tailings as fine aggregate in ultra-high performance concrete[J]. Construction and Building Materials, 2014, 50(2):540-548. https://doi.org/10.1016/j.conbuildmat.2013.10.019

4) Thanks for your kind suggestion very much, and we have added a unit to make the figure clearer (now Line 160).

5) Thanks for your kind suggestion very much, and we have corrected this mistake (now Line 156, 157 and 187).

6) Thanks for your kind suggestion very much. The water reducing agent used in this study is a polycarboxylic-based one, but not AE agent. The air in the slurry is mainly caused by the air introduced during the mixing process and the “water film effect” on the surface of tailings sand.

7) Thanks for your kind suggestion very much. We are very sorry for this basic mistake and we have revised the mistake, which is marked with yellow background in the manuscript (now Line 203).

8) Thanks for your kind suggestion very much. The study is mainly focus on the comparison between tailing sand and river sand. Hence, we use the control variable method. Except the aggregate, the content of other materials should be a certain factor, in order to compare the performance differences of the two aggregates more accurately and carefully. According to our research, the difference between the two aggregates has a significant effect on the fluidity. Therefore, we set the amount of SP as certain content.

9) Thanks for your kind suggestion very much, and we also agree with you that the nested small graph in figure is too much complicated. Therefore, we improved the quantity of graph marked in yellow background (now Line 326).

Thanks for your kind suggestions very much.

Round 2

Reviewer 2 Report

Section 2.4.3: the question still stays, why is the measured shrinkage called autogenous shrinkage? Is it based on any standard? The authors should show their understanding of autogenous shrinkage. If the specimens are not covered to avoid moisture loss, it is not autogenous shrinkage. 

Section 3.5: what does scratch on the aggregate mean? This should be explained better. Where is the scratch on the figure, and the term scratch is also not so scientific. 

Author Response

Question:

Section 2.4.3: the question still stays, why is the measured shrinkage called autogenous shrinkage? Is it based on any standard? The authors should show their understanding of autogenous shrinkage. If the specimens are not covered to avoid moisture loss, it is not autogenous shrinkage. 

Section 3.5: what does scratch on the aggregate mean? This should be explained better. Where is the scratch on the figure, and the term scratch is also not so scientific. 

Answer:

Thanks for your kind suggestion again. According to your suggestion, we have modified the manuscript as below:

Section 2.4.3. The experiment environment of autogenous test is selected to be 20 ± 2°C and a relative humidity of 60 ± 5% referred to the literature (Development of a novel cleaner construction product: Ultra-high performance concrete incorporating lead-zinc tailings, https://doi.org/10.1016/j.jclepro.2018.06.058). At the same time, considering that the maximum particle size of aggregate (0.3mm) is too much finer than that of literature above. Hence, the size of mold (25 mm × 25 mm × 280 mm) referred to the literature (Effect of recycled tyre polymer fibre on engineering properties of sustainable strain hardening geopolymer composites https://doi.org/10.1016/j.cemconcomp.2021.104167) and standard ASTM C490-17. This paragraph has been modified in the manuscript at the same time, with a yellow background.

Section 3.5: Thanks for your kind suggestion. This paragraph has been modified in the manuscript to maker it more clear to be understood, and Figure 14 has also been improved.

Reviewer 3 Report

Thank you very much for addressing all my comments. The paper is recommended for acceptance in its current state. 

Author Response

Answer: Thank you for your affirmation on this study and thank you again for your valuable comments on our manuscript very much.